# Constituents from the Seeds of *Sophora Alopecuroides* L.

**DOI:** 10.3390/molecules25020411

**Published:** 2020-01-19

**Authors:** Zi-Jian Rong, Gao-Sheng Hu, Shi-Yi Lin, Ting Yan, Na Li, Yue Zhao, Jing-Ming Jia, An-Hua Wang

**Affiliations:** 1School of Traditional Chinese Materia Medica, Shenyang Pharmaceutical University, Shenyang 110016, China; rongzijianlamia@gmail.com (Z.-J.R.); hugsh_2011@163.com (G.-S.H.); yantinglingxiao@163.com (T.Y.); ln4311413@163.com (N.L.); zhaoyue82@126.com (Y.Z.); 2College of pharmacy, Shenyang Pharmaceutical University, Shenyang 110016, China; lin1997615@163.com

**Keywords:** *S. alopecuroides* L., isoflavone glucosides, structure elucidation, cytotoxicity

## Abstract

Three new isoflavone glucosides, kudonol A−C (**1**–**3**), two new ester derivatives of phenylpropanoid, kudolignan A and B (**4**–**5**) and five known compounds, (−)-maackiain (**6**), neoliquiritin (**7**), methyl 4-coumarate (**8**), methyl ferulate (**9**) and (+)-wikstromol (**10**), were isolated from an extract of dried seeds of the traditional Chinese medicinal plant *Sophora alopecuroides* L. Their structures were established by NMR and HRESIMS data analyses. The monosaccharide part’s configuration of isoflavone glucosides was confirmed by acid hydrolysis and analyzed by a JAsco OR-4090 chiral detector, comparing it to standard substance D-glucose. The cytotoxicity effects against HeLa, Hep3B, MCF-7 and H1299 cells were tested by CCK-8 assay.

## 1. Introduction

*Sophora alopecuroides* L. is a traditional Chinese herbal plant, known as “Ku-Dou-Zi” in China, and is a perennial herbaceous plant in the Leguminosae family, which is widely distributed in the deserts of northern China, especially in the Xinjiang and Ningxia provinces as well as the Inner Mongolia Autonomous Region. The seeds of *S. alopecuroides* have a long history as a traditional Chinese medicine, utilized for the treatment of eczema, acute pharyngolaryngeal infection, sore throat, acute dysentery, and gastrointestinal hemorrhage [1].

Previous phytochemical investigations on *S. alopecuroides* have led to the isolation of alkaloids, flavonoids, isoflavonoids, and rotenoids [2]. Compounds isolated from the seeds of this plant have been used to treat leukemia [3], trophoblastic tumors and inflammation in traditional Chinese medicine. For example, sophocarpine was found to be effective against tumors, both in cell proliferation and metastasis in liver cancer [4].

During our present exploration for the diterpenoids of *S. alopecurioides*, five previously undescribed compounds were isolated and identified successfully for the first time (Figure 1). The structures of the isolated isoflavone glucosides were determined from various spectroscopic data. Finally, the inhibitory effects of some of the isolates against four cancer cell lines were evaluated in vitro using a CCK-8 bioassay.

## 2. Results and Discussion

Compound **1** was obtained as a yellow powder and exhibited an ion peak at *m*/*z* 591.1512 ([M − H]^−^, calcd 591.1581) in its HRESIMS data, which indicated a molecular formula of C_31_H_28_O_12_. The indices of hydrogen deficiency (IHDs) of compound **1** was 18. The ^13^C NMR data of compound **1** comprised the parent nucleus of an isoflavonoid and a monosaccharide. On the basis of the ^1^H-NMR spectroscopic data (Table 1), six aromatic proton signals at *δ*_H_ 8.03 (1H, d, *J* = 8.4 Hz), *δ*_H_ 7.20 (1H, d, *J* = 2.4 Hz), *δ*_H_ 7.14 (1H, dd, *J* = 8.4, 2.4 Hz), *δ*_H_ 6.95 (1H, d, *J* = 8.4 Hz), *δ*_H_ 7.04 (1H, d, *J* = 1.8 Hz) and *δ*_H_ 6.81 (1H, dd, *J* = 8.4, 1.8 Hz) were observed, which were deduced to be two ABX coupled system in a sub-structure. Meanwhile, fifteen carbon signals at *δ*_C_ 153.2, 123.5, 174.5, 127.0, 115.6, 161.2, 103.4, 156.9, 118.5, 124.4, 116.4, 146.0, 147.6, 112.0 and 119.5 in the ^13^C NMR spectrum (Appendix A) indicated the presence of structural fragments of isoflavones. A spin-spin system at *δ*_H_ 5.18(H-1″)/3.36(H-2″)/3.37(H-3″)/3.27(H-4″)/3.84(H-5″)/4.45(H-6″) was established on the basis of an ^1^H-^1^H COSY experiment, which indicated that there may be a six-carbon sugar in the structure. In its NOESY spectrum, the correlations *δ*_H_ 5.18(H-1″)/3.37(H-3″)/3.84(H-5″) and 3.36(H-2″)/3.27(H-4″)/4.45(H-6″) established the 1″a-H, 3″a-H, 5″a-H in the six-carbon sugar ring. Six carbon signals at *δ*_C_ 99.7, 73.0,76.3, 70.0, 73.9 and 63.3 suggested the existence of glucose as well as an anomeric proton at *δ*_H_ 5.18 (d, *J* = 7.2 Hz). Additionally, an acid hydrolysis experiment performed on compound **1** afforded D-glucose, which was determined using a JAsco OR-4090 detector with D-glucose standard sample. Then, a parahydroxy cinnamic acid was deduced in the structure from the signals at *δ*_H_ 7.52 (2H, d, *J* = 8.4 Hz) and 6.79 (2H, d, *J* = 8.4 Hz) and a pair of trans-ene hydrogen signals at *δ*_H_ 7.56 (1H, d, *J* = 15.6 Hz, 7‴-H) and 6.39 (1H, d, *J* = 15.6 Hz, 8‴-H) in the ^1^H NMR spectrum as well as a carbonyl carbon signal at *δ*_C_ 166.3 in the ^13^C NMR spectrum. The analysis of the spectroscopic data indicated that compound **1** was similar to the known compound calycosin-7-O-D-glucopyranoside [5], and the difference mainly lay in the parahydroxy cinnamic acid group. The exact structure of compound **1** was established by 2D-NMR. In the HMBC spectrum, a methoxide signal at *δ*_H_ 3.80 had a long-range correlation with *δ*_C_ 147.6(C-4′), which indicated a methoxy group connected on the C ring of this isoflavone. A long-range correlation was observed between *δ*_H_ 5.18 (H-1′) and *δ*_C_ 161.2 (C-7), which located the D-glucose at C-7 of the isoflavone, and the anomeric proton at *δ*_H_ 5.18 (d, *J* = 7.2 Hz) also indicated the presence of a β glycosidic bond. The parahydroxy cinnamic acid group was posited to the C-6 of D-glucose by the HMBC correlations of *δ*_H_ 4.45 (H-6″)/*δ*_C_ 166.3 (C-9‴). Based on the abovementioned data, the structure of compound **1** was elucidated as Calycosin-7-O-*β*-D-(6″-hydroxy cinnamate)-glucopyranoside and named as kudonol A.

Compound **2**, a yellow powder, gave the molecular formula C_30_H_26_O_11_ through HRESIMS of its quasi-molecular ion peak at *m*/*z* 561.1405 ([M − H]^−^, calcd 561.1475). The spectroscopic data indicated that compound **2** was similar to compound **1**, except for an absence of a methoxy group and an extra aromatic proton signal. Based on a long-range correlation of H-1″ (*δ*_H_ 5.12, *J* = 7.2 Hz)/C-7(*δ*_C_ 163.1) observed in the HMBC data, the glucose moiety was located at C-7 of the isoflavone fragments. Meanwhile, a structural fragment of cinnamic acid was evidenced by signals at *δ*_H_ 7.55 (1H, d, *J* = 8.4 Hz), 7.35 (1H, d, *J* = 8.4 Hz) and 7.34 (1H, m) in the ^1^H NMR spectrum and a carbonyl carbon signal at *δ*_C_ 168.1 in the ^13^C NMR spectrum. Additionally, a long-range correlation was observed from *δ*_H_ 4.46 (H-6″) to *δ*_C_ 168.1 (C-9‴), which located the cinnamic acid at C-6 of the glucose. Compared with compound **1**, the structure of compound **2** has a hydroxy group instead of the methoxy group present in the B ring of the isoflavone parent nucleus. Consequently, the structure of compound **2** was elucidated to be 7,3′,4′-trihydroxyisoflavone-7-O-*β*-D-(6″-cinnamic acid)-glucopyranoside and named as kudonol B.

Compound **3** had the molecular formula C_28_H_24_O_12_ according to the HRESIMS data from its quasi-molecular ion peak at *m*/*z* 553.1336 ([M + H]^+^, calcd 553.1268). On the basis of spectroscopic data, compound **3** was deduced to be an isoflavone glycoside, which was similar to compound **1**; the main difference was that cinnamic acid was replaced by a 2-hydroxy benzoyl group, as well as a hydroxy group instead of the methoxy at C-4′. The hydrolysis experiment on compound **3** gave a monosaccharide, which was identified to be of the D-glucopyranosyl group by the corresponding standard substances. Through the analysis of spectral data, a salicylic acid group was deduced from *δ*_C_ 113.5, 162.8, 118.4, 137.0, 120.4, 131.2 and 171.0 in the ^13^C NMR spectrum, and that was attached to the C-6 position of the glucose with the carboxyl of itself, which was indicated from the long-correlation between *δ*_H_ 4.81 (H-6″) and *δ*_C_ 171.0 (C-7‴). Consequently, the structure of compound **3** was elucidated to be 7,3′,4′-trihydroxyisoflavone-7-*O*-β-d-(6″-salicylic acid)-glucopyranoside and was named as kudonol C.

Compound **4** was obtained as a pale yellow powder and exhibited an ion peak at *m*/*z* 455.1673 ([M + Na]^+^, calcd 455.1682) in its HRESIMS data, which indicated a molecular formula of C_23_H_28_O_8_. The ^1^H NMR spectrum of compound 4 (see Table 2) showed six aromatic proton signals at *δ*_H_ 7.16 (1H, d, *J* = 1.8 Hz, H-2), 6.93 (1H, d, *J* = 8.4 Hz, H-5), 7.08 (1H, dd, *J* = 8.4, 1.8 Hz, H-6), 6.98 (1H, d, *J* = 1.8 Hz, H-2′), 6.76 (1H, d, *J* = 7.8 Hz, H-5′) and 6.84 (1H, dd, *J* = 7.8, 1.8 Hz, H-6′), revealing the presence of two ABX system aromatic rings. Two olefin proton signals were observed at *δ*_H_ 7.16 (1H, d, *J* = 16.2 Hz, H-7) and 6.40 (1H, d, *J* = 16.2 Hz, H-8) in the lower field region in the ^1^H NMR spectrum. Two oxymethine protons at *δ*_H_ 4.40 (1H, d, *J* = 6.6 Hz, H-7′) and 4.54 (1H, m, H-8′), and two oxymethylene protons at *δ*_H_ 3.86 (2H, d, *J* = 4.8 Hz, H-9′), established the presence of a 1,2,3-propanetriol moiety. Additionally, one ethoxy group at *δ*_H_ 4.26 (2H, q, *J* = 7.2 Hz, H-1″) and 1.35 (3H, t, *J* = 7.2 Hz, H-2″) and three methoxy groups at *δ*_H_ 3.83 (s, 3, OCH_3_), 3.82 (s, 3, OCH_3_) and 3.26 (s, 3, OCH_3_) were observed in the ^1^H NMR spectrum. The ^13^C NMR spectrum of compound **4** (see Table 2) showed 23 carbon signals. Aside from the carbon signal of the ethoxy unit and the three methoxy groups, the remaining 18 carbon signals included a carbonyl signal at *δ*_C_ 169.1 (C-9), double ring olefin carbon signals at *δ*_C_ 116.8 (C-8) and 146.1 (C-9), 12 aromatic carbons, and three aliphatic carbons at 83.7 (C-7′), 84.7 (C-8′) and 62.3 (C-9′). The HMBC correlations of H-7 at *δ*_H_ 7.61 with C-2, of H-8 at *δ*_H_ 6.40 with C-6 and C-9, and of H-7′ at *δ*_H_ 4.40 with C-2′, C-5′, C-6′ and C-8′ confirmed the presence of two phenyl propanoid units. In the HMBC spectrum, the correlation of H-8′ at *δ*_H_ 4.54 with C-4 at *δ*_C_ 151.8 suggested that compound **4** was an 8′-O-4 system neolignan. The methoxy group was determined to be at C-3, C-3′ and C-7′, based on the HMBC correlation of the methoxy groups at *δ*_H_ 3.83 with C-3 at *δ*_C_ 151.8 and at *δ*_H_ 3.82, with C-3′ at *δ*_C_ 148.9 and *δ*_H_ 3.26, and with C-7′ at *δ*_C_ 83.7. Compound **5**, a pale yellow powder, gave the same molecular formula through HRESIMS, the spectroscopic data indicated that compound **5** was similar to compound **4**, except for the difference of their absolute configuration. Regarding the configurations of compounds **4** and **5**, the difference in chemical shifts of both H-9′ protons is the parameter that must be used to establish the relative configuration around the chiral centers H-7′ and H-8′ [6,7], in compound **4** both H-9′ protons seem to be isochronous (*δ*_H_ 3.86), whereas there is a difference of 0.20 ppm (*δ*_H_ 3.71 and 3.51) in the chemical shift of these protons in compound **5**, thus confirming that **4** and **5** have erythro and threo relative configurations, respectively, namely these are an erythro/threo pair. Thus, the structure of compound **4** was determined to be erythro-4′,9′-dihydroxy-3,3′,7′-trimethoxy-9-ethyoxyl-8−4-oxyneolignan and was named as kudolignan A (**4**), and **5** was determined to be threo-4′,9′-dihydroxy-3,3′,7′-trimethoxy-9-ethyoxyl-8−4-oxyneolignan named as kudolignan B (**5**).

The other known compounds were identified as (−)-maackiain (**6**) [8], neoliquiritin (**7**) [9], methyl 4-coumarate (**8**) [10], methyl ferulate (**9**) [11] and (+)-Wikstromol (**10**) [12] from the comparison of their NMR data with that reported in the literature.

Compounds **1**–**6** were tested for their cytotoxicity against four human cancer cell lines (HeLa, Hep3B, MCF-7 and H1299) and one normal human liver cell line (LO2) (see Table 3). Experimental results show that all of the compounds showed no cytotoxicity to the LO2 cell line, compound **2** showed moderate inhibition of cell proliferation against several cancer cell lines, and compound **1** showed moderate inhibition of cell proliferation against Hep3B cells. On the basis of structural analysis, the activity of inhibiting the compounds’ cytotoxicity against tumor cells was probably due to the trans-cinnamic acid moiety.

## 3. Experimental Section

### 3.1. General Experimental Procedures

NMR spectra were run in DMSO-*d*_6_ and CD_3_OD on a Varian Mercury NMR spectrometer. Analytical HPLC data were collected on an Agilent 1260 infinity II instrument (Thermo Scientific dionex). The UV spectra were measured by an Agilent 1260 infinity II UV–vis spectrophotometer in methanol. Preparative HPLC was performed by a Saipuruisi MH-LC 52 instrument with an Elite UV2300 detector and an YMC C18 column (250 × 20 mm, 5 µm). The HRESIMS data were obtained using an Agilent 1290 series 6540 UHD accurate mass Q-TOF mass spectrometer using direction injection. The sugar configurations were determined by a JAsco OR-4090 chiral detector. Column chromatographic separations were carried out on silica gel H-60 (Qingdao Marine Chemical Group Corporation, Qingdao, China), Polyamide (Shanghai Yiyan biology technology Co. Ltd., Zhengzhou, China) and Sephadex LH-20 (Shanghai Yi-He biological technology Co. Ltd., Shanghai, China). The Acetonitrile and Methanol used were of chromatographic grade, and were purchased from Fisher in America. All other solvents used of chemical grade (Da-Mao Chemical Co. Ltd., Tianjin, China).

### 3.2. Plant Materials

The dry seeds of *S. alopecuroides* L. were collected from Alxa League in Inner Mongolia Autonomous Region and identified by Prof. Liu Yong, Jiang-Xi University of Chinese Medicine. The specimens were deposited in the specimen room of traditional Chinese Medicine in Shenyang Pharmaceutical University (SPU-2014-0714-06).

### 3.3. Extraction and Isolation

The dry seeds of *S. alopecuroides* L. (50 kg) were extracted by 70% ethanol aqueous solution, refluxed (1.5 h × 3) and adjusted to a pH of 2 by 1%HCl, and then partitioned with EtOAc. The EtOAc portion (336 g) was partitioned by polyamide chromatography (EtOH/H_2_O) and yielded four fractions (100% water, 30% EtOH, 60% EtOH, 90% EtOH). Afterwards, the fraction of 60% EtOH (51 g) was dealt with by a silica gel column by elution with CH_2_Cl_2_:CH_3_OH (100:1–0:100) in sequence to give fractions 1–19. Sub-fraction 12 was separated by medium pressure ODS CC (0.03% TFA, 45–100% CH_3_OH aqueous) to obtain 22 fractions, 1′–22′. Fraction 6′ was purified by preparative HPLC with CH_3_CN-H_2_O (30%, 0.03% TFA) to get compound **1** (12.7 mg, t_R_ = 16.06 min), compound **2** (18.9 mg, t_R_ = 21.50 min) and compound **3** (20.4 mg, t_R_ = 17.45 min). Fraction 2′ was purified by preparative HPLC with CH_3_CN-H_2_O (15%, 0.03% TFA) to get compound **7** (2.2 mg, t_R_ = 16.06 min). Sub-fraction 4 was separated by Sephadex LH-20 (90% CH_3_OH aqueous) to obtained 16 fractions 1″–16″. Fraction 9″ was purified by preparative HPLC with CH_3_OH–H_2_O (60%, 0.03% TFA) to get compound **4** (3.1 mg, t_R_ = 21.44 min) and compound **5** (3.2 mg, t_R_ = 23.01 min). Sub-fraction 3 was separated by medium pressure ODS CC (0.03% TFA, 10–100% CH_3_OH aqueous) to obtain 14 fractions 1‴–14‴. Fraction 10‴ was purified by preparative HPLC with CH_3_OH–H_2_O (40%, 0.03% TFA) to get compound **8** (2.1 mg, t_R_ = 40.02 min) and compound **9** (2.5 mg, t_R_ = 42.33 min). Fraction 11‴ was purified by preparative HPLC with CH_3_OH–H_2_O (50%, 0.03% TFA) to get compound **6** (42.4 mg, t_R_ = 24.21 min). Fraction 9‴ was purified by preparative HPLC with CH_3_OH–H_2_O 35%, 0.03% TFA) to get compound **10** (2.3 mg, t_R_ = 39.15 min).

#### 3.3.1. Kurosol A (**1**)

A yellow powder; [α]_D_^20^ −39.12(c 0.3, MeOH); UV λ_max_ 196, 264 and 292; ^1^H NMR (DMSO-*d*_6_, 600 MHz) and ^13^C NMR (DMSO-*d*_6_, 150 MHz), see Table 1; HRESIMS *m*/*z* 591.1512 ([M − H]^−^, calcd 591.1581).

#### 3.3.2. Kurosol B (**2**)

A yellow powder; [α]_D_^20^ −71.97(c 0.8, MeOH); UV λ_max_ 200, 218 and 264; ^1^H NMR (CD_3_OD, 600 MHz) and ^13^C NMR (CD_3_OD, 150 MHz), see Table 1; HRESIMS *m*/*z* 561.1405 ([M − H]^−^, calcd 561.1475).

#### 3.3.3. Kurosol C (**3**)

A yellow powder; [α]_D_^20^ −28.00(c 0.8, MeOH); UV λ_max_ 204 and 290; ^1^H NMR (CD_3_OD, 600 MHz) and ^13^C NMR (CD_3_OD, 150 MHz), see Table 1; HRESIMS *m*/*z* 553.1336 ([M + H]^+^, calcd 553.1268).

#### 3.3.4. Kudolignan A (**4**)

A pale yellow powder; [α]_D_^20^ −14.60(c 0.2, MeOH); UV λ_max_ 198, 230 and 324; ^1^H NMR (CD_3_OD, 600 MHz) and ^13^C NMR (CD_3_OD, 150 MHz), see Table 2; HRESIMS *m*/*z* 455.1673 ([M + Na]^+^, calcd 455.1682).

#### 3.3.5. Kudolignan B (**5**)

A pale yellow powder; [α]_D_^20^ −21.53(c 0.1, MeOH); UV λ_max_ 200, 230 and 322; ^1^H NMR (CD_3_OD, 600 MHz) and ^13^C NMR (CD_3_OD, 150 MHz), see Table 2; HRESIMS *m*/*z* 455.1676 ([M + Na]^+^, calcd 455.1682).

### 3.4. Acid Hydrolysis

Five milligrams of powder of compounds **1**, **2** and **3** were dissolved in CH_3_OH (the least amount) and participated in an acid hydrolysis reaction with 2 M of HCl in a 90 °C water bath over 5 hours [13]. The reaction mixture was cooled to room temperature and extracted by EtOAc three times. Then, the water layer was evaporated in vacuo by a rotatory evaporator to remove the remaining EtOAc. The residue was dissolved in water again and next analyzed by HPLC coupled to a JAsco OR-4090 chiral detector, comparing it to standard substance glucose (chromatography column: Shodex Asahipak NH2P-504E, CH_3_CN: H_2_O (3:1), 0.8 mL/min).

### 3.5. Cytotoxicity Assay

Compounds **1**–**6** were tested for their cytotoxicity against the MCF-7, Hela, H1299 and Hep-3B cell lines by means of the CCK-8 method as described in the literature [14,15]. 5-fluorouracil was used as a positive control.

## 4. Conclusions

Three new isoflavone glucosides, termed kudonol A−C (**1**–**3**), two new ester derivatives of phenylpropanoid, kudolignan A and B (**4**–**5**) together with five known compounds (**6**–**10**), were isolated from the seeds of *S. alopecuroides*. The absolute configurations of compounds **4**–**5** were determined by the extensive analysis of spectroscopic data and quantum chemical ECD calculations. Unfortunately, the cytotoxicity of compounds **2** and **6** that were evaluated against the tumor cell lines showed only weak activity. In addition, further biological assays of these compounds and the structural diversity of *S. alopecuroides* are worth exploring in future research.

## Figures and Tables

**Figure 1 molecules-25-00411-f001:**
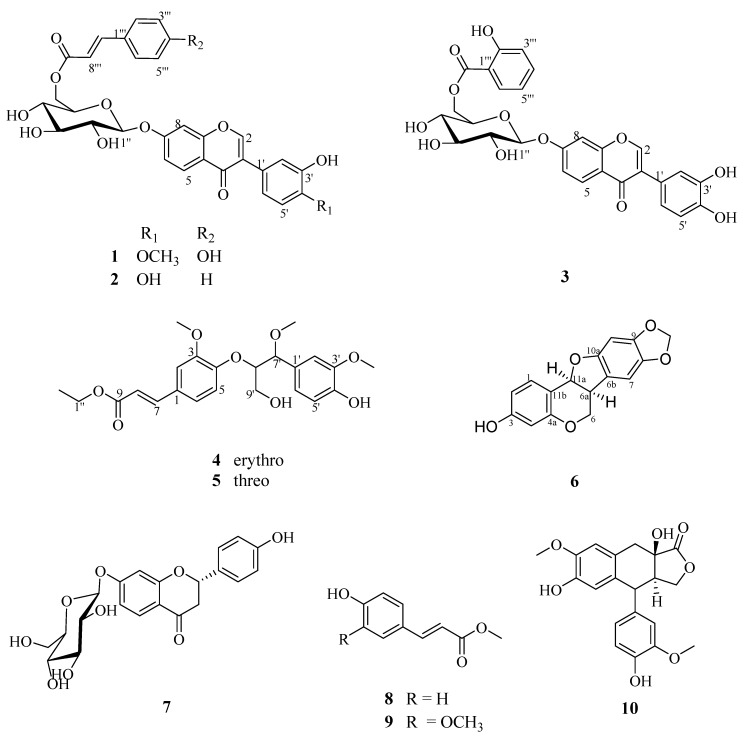
Structures of compounds **1**–**10**.

**Table 1 molecules-25-00411-t001:** ^1^H NMR data (600 MHz) and ^13^C NMR data (150 MHz) of compounds **1**, **2** and **3**.

Position	1(DMSO-*d*_6_)		2(CD_3_OD)		Position	3(CD_3_OD)	
	*δ*_H_ (*J* in Hz)	*δ* _C_	*δ*_H_ (*J* in Hz)	*δ* _C_		*δ*_H_ (*J* in Hz)	*δ* _C_
2	8.03, s	153.2	7.50, s	154.7	2	8.04, s	154.9
3	-	123.5	-	124.5	3	-	124.6
4	-	174.5	-	177.8	4	-	178.0
5	8.03, d (8.4)	127.0	8.09, d (9.0)	128.2	5	8.08, d (8.4)	128.3
6	7.14, dd (8.4, 2.4)	115.6	7.16, dd (9.0, 1.8)	117.5	6	7.19, dd (8.4, 2.4)	117.0
7	-	161.2	-	163.1	7	-	163.0
8	7.20, d (2.4)	103.4	7.15, d (1.8)	104.5	8	7.18, d (2.4)	104.8
9	-	156.9	-	158.9	9	-	159.0
10	-	118.5	-	120.2	10	-	120.2
1′	-	124.4	-	126.0	1′	-	126.2
2′	7.04, d (1.8)	116.4	6.93, d (1.8)	117.4	2′	7.06, s	117.5
3′	-	146.0	-	146.1	3′	-	146.2
4′	-	147.6	-	146.7	4′	-	146.8
5′	6.95, d (8.4)	112.0	6.82, d (7.8)	116.2	5′	6.94, m	116.4
6′	6.81, dd (8.4, 1.8)	119.5	6.67, dd (7.8, 1.8)	121.6	6′	6.94, m	121.7
1″	5.18, d (7.2)	99.7	5.12, d (7.2)	101.5	1″	5.21, d (7.8)	101.2
2″	3.36, m	73.0	3.60, m	74.7	2″	3.62, m	74.6
3″	3.37, m	76.3	3.60, m	77.8	3″	3.62, m	77.8
4″	3.27, m	70.0	3.47, m	72.0	4″	3.51, m	72.0
5″	3.84, m	73.9	3.92, m	75.7	5″	4.00, m	75.5
6″	4.45, dd (12.0, 1.8) 4.21, m	63.3	4.64, dd (12.0, 1.8) 4.45, m	64.9	6″	4.81, dd (12.0, 1.8) 4.56, m	65.6
1‴	-	125.0	-	135.7	1‴	-	113.5
2‴/6‴	7.52, d (8.4)	130.3	7.55, d (8.4)	129.2	2‴	-	162.8
3‴/5‴	6.79, d (8.4)	115.8	7.35, d (8.4)	130.2	3‴	6.94, m	118.4
4‴	-	159.9	7.34, m	131.7	4‴	7.52, m	137.0
7‴	7.56, d (15.6)	144.9	7.70, d (15.6)	146.6	5‴	6.94, m	120.4
8‴	6.39, d (15.6)	113.9	6.56, d (15.6)	118.7	6‴	7.95, dd (7.8, 1.8)	131.2
9‴	-	166.3	-	168.1	7‴	-	171.0
-OCH_3_	3.80, s	55.6	-	-	-	-	-

**Table 2 molecules-25-00411-t002:** ^1^H NMR data (600 MHz) and ^13^C NMR data (150 MHz) of compounds **4**–**5** (CD_3_OD).

Position	4		5	
	*δ*_H_ (*J* in Hz)	*δ* _C_	*δ*_H_ (*J* in Hz)	*δ* _C_
1	-	129.5	-	129.4
2	7.16, d (1.8)	112.3	7.24, d (2.4)	112.3
3	-	151.8	-	152.4
4	-	151.7	-	151.7
5	6.93, d (8.4)	117.4	7.04, d (8.4)	117.2
6	7.08, dd (8.4, 1.8)	123.3	7.13, dd (8.4, 2.4)	123.5
7	7.61, d (16.2)	146.1	7.64, d (16.2)	146.1
8	6.40, d (16.2)	116.8	6.42, d (16.2)	116.8
9	-	169.1	-	169.1
1′	-	130.6	-	130.8
2′	6.98, d (1.8)	112.5	7.00, d (1.8)	112.1
3′	-	148.9	-	149.1
4′	-	147.5	-	147.6
5′	6.76, d (7.8)	115.6	6.81, d (7.8)	116.0
6′	6.84, dd (7.8, 1.8)	122.3	6.85, dd (7.8, 1.8)	121.6
7′	4.40, d (6.6)	83.7	4.47, d (6.0)	84.2
8′	4.54, m	84.7	4.50, m	85.1
9′	3.86, d (4.8)	62.3	3.71, dd (12.0, 4.2) 3.51, q (6.0)	62.2
1″	4.26, q (7.2)	61.5	4.26, q (7.2)	61.5
2″	1.35, t (7.2)	14.6	1.35, t (7.2)	14.6
3-OCH_3_	3.83, s	56.5	3.92, s	56.6
3′-OCH_3_	3.82, s	56.3	3.85, s	56.3
7′-OCH_3_	3.26, s	57.0	3.26, s	57.2

**Table 3 molecules-25-00411-t003:** Inhibition effects on the growth of tumor cells in vitro (IC_50_, μM).

Compounds	IC_50_ ± SEM (μM)				
	HeLa	Hep3B	MCF-7	H1299	LO2
**1**	118.237 ± 15.524	> 150	> 150	> 150	> 150
**2**	68.033 ± 8.321	85.366 ± 27.313	> 150	77.366 ± 17.309	> 150
**3**	> 150	> 150	> 150	> 150	> 150
**4**	> 150	> 150	> 150	> 150	> 150
**5**	> 150	> 150	> 150	> 150	> 150
**6**	97.590 ± 20.504	124.909 ± 35.021	99.742 ± 17.001	68.376 ± 11.528	> 150
**5-Fu**	15.990 ± 0.121	32.223 ± 3.257	14.450 ± 2.193	24.450 ± 6.153	> 150

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
