# Peer review of "Constituents from the Seeds of Sophora Alopecuroides L."

_molecules, 2020, doi:10.3390/molecules25020411_

Round 1
Reviewer 1 Report
The article describes the isolation and cytotoxic activities of several new isoflavone glucosides and phenylpropanoid esters from extracts of Sophora alopecuroides seeds.
Although it seems to contain some valuable information, a major revision is required to generate a new version of the manuscript that might be acceptable for publication. More precisely, the following aspects should be addressed:
1) The english language employed is rather poor, making some parts of the manuscript difficult to understand. The manuscript must therefore be ckecked by a native speaker prior to the submission of any revised version.
2) A more detailed explanation of the experiments performed to determine the absolute configuration of the sugar residue in compounds 1-3 should be provided. The identity of glucose as the residue present in the molecules is not fully clear based on the NMR information provided and it can for sure be confirmed via NOESY or ROESY experiments. Additionally, according to paragraph 3.4, it seems that chromatography combined with the use of a chiral detector has been used to assign the sugar as D-glucose, but no evidence of the results is shown in the article. Ideally, the supplementary information should include a chormatogram of the glucose standard and that of the hydrolization product. Additionally, what is the reason behind using fructose as a standard? What about using other monosaccharides instead of fructose?
3) The absolute configuration proposed for compounds 4 and 5 seems to not be fully supported by the experimental evidences provided. According to the information provided on pages 4 and 5, coupling constants of 6.6 and 6.0 between H-7' and H-8' would render erythro and threo relative configurations, respectively. According to reference 6, this small difference is not enough to distinguish between both diastereoisomers and the chemical shifts of both H-9' protons must also be considered to assign the relative configuration in molecules of this structural class. Regarding the absolute configuration of the molecules, the authors propose 8'S and 8'R configurations for compounds 4 and 5, respectively, based on the positive or negative cotton effect observed in their CD spectra at 229 and 228 nm. Please note than in reference 7 the CD spectra shown for four diastereoisomers of this type are very similar, with the only major difference being in the area round 230 nm, whereas the two spectra shown for compounds 4 and 5 in figure 2 of this article seem to belong to structurally different molecules. What is the interpretation of the authors about the positive cotton effects ovserved around 220 and 235 nm in compound 5? In the opinion of this reviewer, a heterenuclear coupling constant analysis should also be performed around carbons C-7' and C-8' to secure the stereochemical proposal. Additionally, perhaps an analysis of NOESY correlations and cotton effects similar to that described in https://www.sciencedirect.com/science/article/pii/S0031942211002731?via%3Dihub would be necessary to secure their stereochemical proposal.
4) Please avoid the use of anticancer or antitumour compounds when only in vitro cytotoxicity against cancer cell lines has been tested.
Author Response
1) The english language employed is rather poor, making some parts of the manuscript difficult to understand. The manuscript must therefore be ckecked by a native speaker prior to the submission of any revised version.
R: Thank you. The whole manuscript has been checked by a native speaker.
2) A more detailed explanation of the experiments performed to determine the absolute configuration of the sugar residue in compounds 1-3 should be provided. The identity of glucose as the residue present in the molecules is not fully clear based on the NMR information provided and it can for sure be confirmed via NOESY or ROESY experiments. Additionally, according to paragraph 3.4, it seems that chromatography combined with the use of a chiral detector has been used to assign the sugar as D-glucose, but no evidence of the results is shown in the article. Ideally, the supplementary information should include a chormatogram of the glucose standard and that of the hydrolization product. Additionally, what is the reason behind using fructose as a standard? What about using other monosaccharides instead of fructose?
R: Thank you very much for your guidance, this section has been revised and added in this article detailly. The problem of glucose relative configuration is a writing error in table 1, the peak is a double peak with a coupling constant of 7.8 Hz, as shown in the figure S13, supplementary material. Additionally, we have added their 1H-1H COSY spectra and chromatogram of the glucose standard (no fructose, it's a lapsus calami) and hydrolyzation products of compounds 1~3 in the supplementary materials (figure S7, S14 and S22), and analyzed the configuration of sugar in detail in the manuscript.
3) The absolute configuration proposed for compounds 4 and 5 seems to not be fully supported by the experimental evidences provided. According to the information provided on pages 4 and 5, coupling constants of 6.6 and 6.0 between H-7' and H-8' would render erythro and threo relative configurations, respectively. According to reference 6, this small difference is not enough to distinguish between both diastereoisomers and the chemical shifts of both H-9' protons must also be considered to assign the relative configuration in molecules of this structural class. Regarding the absolute configuration of the molecules, the authors propose 8'S and 8'R configurations for compounds 4 and 5, respectively, based on the positive or negative cotton effect observed in their CD spectra at 229 and 228 nm. Please note than in reference 7 the CD spectra shown for four diastereoisomers of this type are very similar, with the only major difference being in the area round 230 nm, whereas the two spectra shown for compounds 4 and5 in figure 2 of this article seem to belong to structurally different molecules. What is the interpretation of the authors about the positive cotton effects ovserved around 220 and 235 nm in compound 5? In the opinion of this reviewer, a heterenuclear coupling constant analysis should also be performed around carbons C-7' and C-8' to secure the stereochemical proposal. Additionally, perhaps an analysis of NOESY correlations and cotton effects similar to that described in https://www.sciencedirect.com/science/article/pii/S0031942211002731?via%3Dihub would be necessary to secure their stereochemical proposal.
R: Thank you very much for your guidance. The erythro configuration between two chiral centers at H-7' and H-8' positions was determined by its small coupling constant in these two compounds, there is no difference each other in this article. However, I regret to explain the positive cotton effects observed around 220 and 235 nm in compound 5 in an imperfect way, because there is no significant difference in the NOESY spectra of compounds 4 and 5, as shown in the figure S27 and figure S33 (newly added), supplementary material. Given this situation, is it necessary to delete compounds 4 and 5 from the manuscript for further study in the future? I look forward to getting an opportunity to hear from you,thank you very much.
4) Please avoid the use of anticancer or antitumour compounds when only in vitro cytotoxicity against cancer cell lines has been tested.
R: Thank you. modifications have been made in whole article, including in the title of article.

Reviewer 2 Report
[1] Abstrcat: The first alphabet in compounds names should be small in middle of sentence (Kudonol, Kudolignan, (+)-Wikstromol). Please also check this in whole manuscript.
[2] Introduction: Line 8: „Extracts or monomer isolated from“ How extracts are isolated? Please rephrase.
[3] Page 2, line 2 from bottom: „and a carbonyl carbon signal at δ C 99.7 in the 13 C NMR spectrum“ Please check the carbonyl value and its 166.3???
[4] Page 2, lines 3-4 from bottom: „Then, a parahydroxy cinnamic acid was deduced in the structure by the signals at δ H 7.52 (2H, d, J = 8.4 Hz) and 6.79 (2H, d, J = 8.4 Hz)“ Please mentioned the important H-7''' and H-8''' proton signals.
[5] Page 4, line 12: compound 16???? Or compound 3???
[6] Page 4, Lines 11-20: „isoflavone glycosides, which was similar to 1 and 2, except for an absence of exocyclic ethylenic group.“ What is exocyclic ethylenic group? How compounds 1 and 2 are different from 3 based on exocyclic ethylenic group?. The difference is that cinnamic acid is replaced by 2-hydroxy benzoyl group and please discuss this in detail. Secondly in Table 1, the animeric proton is multiplet and how authors confirmed that the glucose linkage is beta because there is no coupling constant.
[7] Page 5, line 1: Positive HRESIMS data of 2 indicated??? Is it compound 2??
[8] Page 5, lines 1-2, from bottom: „The other compounds were identified as (-)-maackiain (6) [8], neoliquiritin (7) [9], methyl 4-coumarate (8) [10], methyl ferulate (9) [11] and (+)-Wikstromol (10) [9].“ Please metion that these are known compounds and identified with comparison with literature data.
[10] Page 6, General experimental procedures: Please check „CH3OH-d4“ and (MeOH-d4).
[11] Plase use one time the plant name „Sophora alopecuroides L“ and then only „S. alopecuroides“.
[12] Authors should „Conclusion“ part.
Author Response
Q [1] Abstrcat: The first alphabet in compounds names should be small in middle of sentence (Kudonol, Kudolignan, (+)-Wikstromol). Please also check this in whole manuscript.
R: The whole text has been revised, thank you.
Q [2] Introduction: Line 8: „Extracts or monomer isolated from“ How extracts are isolated? Please rephrase.
R: This is a typo, and modifications have been made in this article.
Q [3] Page 2, line 2 from bottom: „ and a carbonyl carbon signal at δ C 99.7 in the 13 C NMR spectrum“ Please check the carbonyl value and its 166.3???
R: This is a spelling error, Carbonyl carbon is at δC 166.3, which has been modified in this article.
Q [4] Page 2, lines 3-4 from bottom: „Then, a parahydroxy cinnamic acid was deduced in the structure by the signals at δ H 7.52 (2H, d, J = 8.4 Hz) and 6.79 (2H, d, J = 8.4 Hz)“ Please mentioned the important H-7''' and H-8''' proton signals.
R: Thank you very much for your guidance, this section has been added in this article.
Q [5] Page 4, line 12: compound 16???? Or compound 3???
R: This is a typo, and modifications have been made in this article.
Q [6] Page 4, Lines 11-20: „isoflavone glycosides, which was similar to 1 and 2, except for an absence of exocyclic ethylenic group.“ What is exocyclic ethylenic group? How compounds 1 and 2 are different from 3 based on exocyclic ethylenic group?. The difference is that cinnamic acid is replaced by 2-hydroxy benzoyl group and please discuss this in detail. Secondly in Table 1, the animeric proton is multiplet and how authors confirmed that the glucose linkage is beta because there is no coupling constant.
R: Thank you very much for your guidance, this section has been revised and added in this article detailly. The problem of glucose relative configuration is a writing error in table 1, the peak is a double peak with a coupling constant of 7.8 Hz, as shown in the Figure S15, supplementary material.
Q [7] Page 5, line 1: Positive HRESIMS data of 2 indicated??? Is it compound 2??
R: It is compound 5. Thank you.
Q [8] Page 5, lines 1-2, from bottom: „The other compounds were identified as (-)-maackiain (6) [8], neoliquiritin (7) [9], methyl 4-coumarate (8) [10], methyl ferulate (9) [11] and (+)-Wikstromol (10) [9].“ Please metion that these are known compounds and identified with comparison with literature data.
R: The NMR data of all known compounds were compared with the data in the literature, meanwhile, the supplementary explanation about them were also carried in the article.
Q [10] Page 6, General experimental procedures: Please check „CH3OH-d4“ and (MeOH-d4).
R: modifications have been made in this article.
Q [11] Plase use one time the plant name „Sophora alopecuroides L“ and then only „S. alopecuroides“.
R: Thank you. modifications have been made in whole article.
Q [12] Authors should „Conclusion“ part.
R: The conclusion has been added in the fourth part of the article.

Round 2
Reviewer 1 Report
This revised version of the manuscript shows some improvement of the language and some additional essential information, but I still feel that my major concerns have not been properly addressed and that some parts of the document need further revision.
Regarding the identity of the glucose moiety in compounds 1-3, I think that the identity of this residue is not clearly confirmed in the article, that it can be established based on NMR evidences and that the explanation on how the NMR spectra confirm the residue as glucose must be included in the manuscript. The NOESY spectrum of compound 3 included in the supplementary information (Fig S20) seems to display correlations between the anomeric proton H1'' and protons H3'' and H5'' confirming that all these protons are axially oriented and on the same face of the molecule. For sure a similar correlation can be observed between protons H2'' and H4'', confirming thus the identity of the residue as glucose. Its D configuration is then confirmed with the HPLC/OD experiment performed. Please note that the identity of the residue as glucose cannot solely be established with the latter experiment as other sugar residues might have the same retention time as glucose in the chromatographic conditions employed and therefore, NMR evidences must also be provided.
Regarding the erythro and threo configurations of compounds 4 and 5, as commented in my previous report, the difference in chemical shifts of both H9' protons is the parameter that must be used to establish the relative configuration around the chiral centers H7' and H8'. As commented in my previous report, reference 6 states that for this kind of structures, small differences in the chemical shift of both H9' protons indicate the existence of an erythro isomer whereas wider differences are associated to the presence of threo isomers. In compound 4 both H9' protons seem to be isocronous, whereas there is a difference of 0.20 ppm in the chemical shift of these protons in compound 5, thus confirming that 4 and 5 have erythro and threo relative configurations, respectively. Regarding the absolute configuration at C-8', it can be proposed taking into account the positive or negative cotton effect at 228 nm, but stating as well that the CD spectra are not clear enough and that this proposal might need further confirmation.
Other minor changes suggested:
Line 13: Replace acquired from the traditional Chinese.. by isolated form an extract of dried seeds of the traditional Chinese...
Line 18: Delete the last part of the last sentence of the abstract, i.e. ...might have antineoplastic potential. The cytotoxicity against tumoral cell lines measured for compound 2 does not confirm at all such potential.
Please change MeOH-d4 by CD3OD all through the text
Sophora alopecurioides should be written in full the first time it appears in the asbtract (line 13) and in the full text (line 23).
Line 30: Consider replacing Monomers by Compounds
Line 34: Replace E. ebracteolata by S. alopecurioides
The word compound/s should be only capitalized at the begginning of a sentence
Line 57: Replace a D-glucose by D-glucose and add the word detector after OR-4090
Line 67: Replace to C-7 by at C-7
Line 68: Replace indicated the by indicated the presence of a
Line 78: Replace indicated by evidenced by signals
Line 80: Replace the long-range by a long-range
Line 81: Replace to C-6 by at C-6
Lines 82-83: Replace is substituted by hydroxyl group on by has a hydroxy group instead of the methoxy group present in
Line 89: Replace by 2-hydroxy by by a 2-hydroxy
Line 90: Replace a methoxy linked at C-4' into a hydroxy by a hydroxy group instead of the methoxy at C-4'
Line 104: Replace shown by observed
Line 105: Replace at the lower by in the lower
Lines 106-107: Replace one oxymethylene proton by two oxymethylene protons
Lines 127-145: Please simplify this paragraph describing compound 5 highligthing only the major differences observed in its NMR spectra and those compound 4
Lines 153 and 154: Consider replacing moderate activity by weak actitivity. IC50 values above 50 μM should not be considered moderate in the opinion of this reviewer.
Line 160: Delete tests
Line 163: Replace The preparative by Preparative
Line 165: Are the authors sure that HRESIMS was performed on the ion trap instrument mentioned?
Line 166: Replace detected by determined and add chiral detector after OR-4090
Line 189: Please indicate the Sephadex LH-20 chromatographic conditions used
Line 220: Replace in the 90 by in a 90
Line 223: Replace analyzed by by analyzed by HPLC coupled to
Lines 228-229: Replace as shown in the previous and acknowledged methos, which has been widely used by as described
Line 229: Delete meanwhile
Lines 236-237: Please revise this sentence
Lines 248-257: I do not consider this abbreviations list is mandatory in Molecules. Most of the terms abbreviated will be for sure understood by potential readers of the article.
DOI must be included in the references where available according to the Instructions for Authors of the journal
Author Response
Dear reviewer,
I have revised the manuscript again according to your opinion. Thank you for your review and your valuable comments on my manuscript.
Kind regards,
An-Hua Wang
The answers to this review are as follows:
Regarding the identity of the glucose moiety in compounds 1-3, I think that the identity of this residue is not clearly confirmed in the article, that it can be established based on NMR evidences and that the explanation on how the NMR spectra confirm the residue as glucose must be included in the manuscript. The NOESY spectrum of compound 3included in the supplementary information (Fig S20) seems to display correlations between the anomeric proton H1'' and protons H3'' and H5'' confirming that all these protons are axially oriented and on the same face of the molecule. For sure a similar correlation can be observed between protons H2'' and H4'', confirming thus the identity of the residue as glucose. Its D configuration is then confirmed with the HPLC/OD experiment performed. Please note that the identity of the residue as glucose cannot solely be established with the latter experiment as other sugar residues might have the same retention time as glucose in the chromatographic conditions employed and therefore, NMR evidences must also be provided.It was added in the manuscript, and all NOESY spectra were shown in the Supplementary data. In the NOESY spectrum of 1, the correlations δH 5.18(H-1'')/ 3.37(H-3'')/ 3.84(H-5'') and 3.36(H-2'')/ 3.27(H-4'')/ 4.45(H-6'') established the 1''a-H, 3''a-H, 5''a-H in the six-carbon sugar ring. Additionally, the correlations δH 5.12(H-1'')/ 3.60(H-3'')/ 3.92(H-5'') and 3.60(H-2'')/ 3.46(H-4'')/ 4.64(H-6'') in the NOESY spectrum of 2; the correlations δH 5.21(H-1'')/ 3.62(H-3'')/ 4.00(H-5'') and 3.62(H-2'')/ 3.51(H-4'')/ 4.56(H-6'') in the NOESY spectrum of 3.
Regarding the erythro and threo configurations of compounds 4and 5, as commented in my previous report, the difference in chemical shifts of both H9' protons is the parameter that must be used to establish the relative configuration around the chiral centers H7' and H8'. As commented in my previous report, reference 6 states that for this kind of structures, small differences in the chemical shift of both H9' protons indicate the existence of an erythro isomer whereas wider differences are associated to the presence of threo isomers. In compound 4 both H9' protons seem to be isocronous, whereas there is a difference of 0.20 ppm in the chemical shift of these protons in compound 5, thus confirming that 4 and 5 have erythro and threo relative configurations, respectively. Regarding the absolute configuration at C-8', it can be proposed taking into account the positive or negative cotton effect at 228 nm, but stating as well that the CD spectra are not clear enough and that this proposal might need further confirmation.The relevant revisions have been reflected in the article. Thank you again for your guidance, which has benefited me a lot.
Other minor changes suggested: Line 13: Replace acquired from the traditional Chinese. by isolated form an extract of dried seeds of the traditional Chinese...It has been changed. Thank you.
Line 18: Delete the last part of the last sentence of the abstract, i.e. ...might have antineoplastic potential. The cytotoxicity against tumoral cell lines measured for compound 2does not confirm at all such potential.It's been removed, thanks for your reminding.
Please change MeOH-d4by CD3OD all through the textThey have been changed in this manuscript and supplementary data.
Sophora alopecurioidesshould be written in full the first time it appears in the asbtract (line 13) and in the full text (line 23).They have been changed in both places.
Line 30: Consider replacing Monomers by CompoundsIt's been replaced, thanks.
Line 34: Replace ebracteolataby S. alopecurioidesThis is a slip of the pen and It's been replaced, thanks.
The word compound/s should be only capitalized at the begginning of a sentenceThey have been changed in whole manuscript.
Line 57: Replace a D-glucose by D-glucose and add the word detector after OR-4090It has been changed. Thank you.
Line 67: Replace to C-7 by at C-7It has been changed. Thank you.
Line 68: Replace indicated the by indicated the presence of aIt's been replaced, thanks.
Line 78: Replace indicated by evidenced by signalsIt's been replaced, thanks.
Line 80: Replace the long-range by a long-rangeIt has been changed. Thank you.
Line 81: Replace to C-6 by at C-6It has been changed. Thank you.
Lines 82-83: Replace is substituted by hydroxyl group on by has a hydroxy group instead of the methoxy group present inIt's been replaced, thanks.
Line 89: Replace by 2-hydroxy by by a 2-hydroxyIt's been replaced, thanks.
Line 90: Replace a methoxy linked at C-4' into a hydroxy by a hydroxy group instead of the methoxy at C-4'It's been replaced, thanks.
Line 104: Replace shown by observedIt's been replaced, thanks.
Line 105: Replace at the lower by in the lowerIt's been replaced, thanks.
Lines 106-107: Replace one oxymethylene proton by two oxymethylene protonsIt's been replaced, thanks.
Lines 127-145: Please simplify this paragraph describing compound 5highligthing only the major differences observed in its NMR spectra and those compound 4It has been changed. Thank you.
Lines 153 and 154: Consider replacing moderate activity by weak IC50values above 50 μM should not be considered moderate in the opinion of this reviewer.It's been replaced, thanks.
Line 160: Delete testsIt's been deleted, thanks.
Line 163: Replace The preparative by PreparativeIt's been deleted, thanks.
Line 165: Are the authors sure that HRESIMS was performed on the ion trap instrument mentioned?The HRESIMS data were obtained using an Agilent 1290 series 6540 UHD accurate mass Q-TOF mass spectrometer, which was been communicated with our test platform further.
Line 166: Replace detected by determined and add chiral detector after OR-4090It has been changed. Thank you.
Line 189: Please indicate the Sephadex LH-20 chromatographic conditions used90% CH3OH aqueous, it was added
Line 220: Replace in the 90 by in a 90It has been changed. Thank you.
Line 223: Replace analyzed by by analyzed by HPLC coupled toIt has been changed. Thank you.
Lines 228-229: Replace as shown in the previous and acknowledged methos, which has been widely used by as describedIt has been changed. Thank you.
Line 229: Delete meanwhileIt has been deleted. Thank you.
Lines 236-237: Please revise this sentenceIt has been revised. Thank you.
Lines 248-257: I do not consider this abbreviations list is mandatory in Molecules. Most of the terms abbreviated will be for sure understood by potential readers of the article.It was a result from the language modification experts, and I couldn't agree with you more and I have deleted it in the manuscript.
DOI must be included in the references where available according to the Instructions for Authors of the journalthey have been added. Thank you.

Reviewer 2 Report
Authors have revised the manuscript and should be accepted now
Author Response
Dear reviewer,
I have revised the manuscript again according to your opinion. Thank you for your review and your valuable comments on my manuscript.
Kind regards,
An-Hua Wang